# Associations with Blood Lead and Urinary Cadmium Concentrations in Relation to Mortality in the US Population: A Causal Survival Analysis with G-Computation

**DOI:** 10.3390/toxics11020133

**Published:** 2023-01-29

**Authors:** Nasser Laouali, Tarik Benmarhnia, Bruce P. Lanphear, Youssef Oulhote

**Affiliations:** 1Department of Biostatistics and Epidemiology, School of Public Health and Health Sciences, University of Massachusetts at Amherst, Amherst, MA 01003, USA; 2Scripps Institution of Oceanography, University of California, San Diego, CA 95616, USA; 3CESP UMR1018, Université Paris-Saclay, UVSQ, Inserm, Gustave Roussy, 94800 Villejuif, France; 4Child and Family Research Institute, BC Children’s Hospital, Vancouver, BC V6H 3N1, Canada; 5Faculty of Health Sciences, Simon Fraser University, Burnaby, BC V5A 1S6, Canada

**Keywords:** metals, cardiovascular, cancer, epidemiology, NHANES

## Abstract

Using the parametric g-formula, we estimated the 27-year risk of all-cause and specific causes of mortality under different potential interventions for blood lead (BLLs) and urinary cadmium (UCd) levels. We used data on 14,311 adults aged ≥20 years enrolled in the NHANES-III between 1988 and 1994 and followed up through 31 Dec 31 2015. Time and cause of death were determined from the National Death Index records. We used the parametric g-formula with pooled logistic regression models to estimate the relative and absolute risk of all-cause, cardiovascular, and cancer mortality under different potential threshold interventions for BLLs and UCd concentrations. Median follow-up was 22.5 years. A total of 5167 (36%) participants died by the end of the study, including 1550 from cardiovascular diseases and 1135 from cancer. Increases in BLLs and creatinine-corrected UCd levels from the 5th to the 95th percentiles were associated with risk differences of 4.17% (1.54 to 8.77) and 6.22% (4.51 to 12.00) for all-cause mortality, 1.52% (0.09 to 3.74) and 1.06% (−0.57 to 3.50) for cardiovascular disease mortality, and 1.32% (−0.09 to 3.67) and 0.64% (−0.98 to 2.80) for cancer mortality, respectively. Interventions to reduce historical exposures to lead and cadmium may have prevented premature deaths, especially from cardiovascular disease.

## 1. Introduction

Cardiovascular disease and cancer are the two leading causes of death, accounting for 17.9 million and 9 million deaths worldwide, respectively, in 2016 [1]. While there has been a marked decline in cardiovascular disease mortality rates, cancer mortality is increasing and is even the leading cause of death in many countries [2]. In 2021, it was estimated that 608,570 Americans would die from cancer, corresponding to more than 1600 deaths per day [3]. These two causes of death, while seemingly different, share common risk factors, including being overweight or obese [4,5], not being physically active [6,7], having a poor diet [8], and exposure to environmental chemicals such as lead and cadmium [9].

Lead and cadmium are among the most common environmental and occupational pollutants derived from natural resources or as a byproduct of industries such as mining. In the last decades, lead and cadmium exposures have declined sharply in the United States [9,10]; however, in 2019, the US Agency for Toxic Substances and Disease Registry ranked lead and cadmium second and seventh, respectively, on the National Priorities List of substances that pose the most significant potential threat to human health [11]. 

While epidemiological studies have consistently reported that exposures to lead and cadmium above the safe standards are associated with increased risks of cardiovascular, cancer, and all-cause mortality [12,13], studies have also reported that chronic low-concentration lead exposure (1–5 μg/dL in the blood) may increase the risk of premature death, especially from cardiovascular diseases [14,15]. In contrast, there are conflicted results for cancer mortality risk at this range of exposure [15]. Results on urinary cadmium concentrations and mortality were reviewed by Larsson et al. in a recent meta-analysis, which highlighted the limited number of prospective studies (*n* = 9 studies) and conflicting results, concluding that there is a need for further large prospective studies [16]. Some limitations persist when analyzing the prospective associations of lead and cadmium with mortality outcomes. First, the non-adjustment for dietary intake (one of the main sources of exposure) and predictors of cancer and cardiovascular disease mortality. Second, the non-consideration of mutual adjustment between these two metals that could confound each other. The two metals share common sources and may also share common mechanistic pathways, such as inflammation, oxidative stress, and recently, epigenetic changes [17]. Third, all previous prospective studies used a Cox proportional hazard model to estimate a single averaged hazard ratio, although the hazard ratio may change over the study’s follow-up. As a result, the conclusions from the study may critically depend on the duration of the follow-up [18]. Finally, previous studies focused on the consequences of exposures to these metals, but it is also important to model the population level impact of potential policies or interventions aimed at reducing exposure levels for informing public health. The hazard ratio alone does not provide this information since the clinical significance depends on the baseline rate [19]. Thus, it is necessary to address all of these limitations and consider the use of models that allow us to overcome the proportional hazards assumption and evaluate the impact of potential interventions on these exposures.

Therefore, we first aimed to extend the follow-up of previous studies in the US National Health and Nutrition Examination Survey (NHANES-III). Secondly, we aimed to use a pooled logistic regression model for the discrete-time hazard within the parametric g-formula to address previous limitations while flexibly simulating the expected all-cause and specific causes of mortality distributions for hypothetical interventions related to lead and cadmium exposures in the US population.

## 2. Methods

### 2.1. Design and Participants

The US National Health and Nutrition Examination Survey (NHANES) data, which were collected, stored, and analyzed by the Centers for Disease Control and Prevention (CDC), were used for this study. The NHANES is an ongoing survey conducted by the CDC that uses a representative sample of non-institutionalized civilians in the US, selected using a complex, multistage, stratified, clustered probability design. Information on participants was collected by interviews and in personal physical examinations by the CDC. The interview includes background information such as socio-demographic, dietary, and health-related questions. The examination component consists of medical and physiological measurements, as well as laboratory tests. The National Center for Health Statistics Ethics Review Board approved all NHANES protocols, and all survey participants completed a consent form. The detailed protocol on NHANES methodology and data collection is available at https://www.cdc.gov/nchs/nhanes/index.htm (accessed on 12 March 2021). For this study, adults aged ≥20 years enrolled in the NHANES-III between 1988 and 1994 with data on blood lead and urinary cadmium concentrations (*n* = 16,040) were included. Exposure and covariate data from NHANES-III were then linked to the National Death Index mortality data.

### 2.2. Mortality Data

A full description of the mortality linkage method is available from the National Center for Health Statistics (NCHS) [20]. Briefly, the de-identified and anonymized data of the NHANES III participants were linked to National Death Index mortality files based on 12 identifiers for each participant (e.g., Social Security number, sex, and date of birth) with a probabilistic matching algorithm to determine mortality status. The NCHS public-use linked mortality file provides mortality follow-up data from the date of NHANES III survey participation until 31 December 2015 (1988–2015). Participants with no matched death record at this date were assumed to be alive during the entire follow-up period. In a validation study using mortality-linked data from the first NHANES study (NHANES-I; 1971–75), 96% of deceased participants and 99% of those still alive were classified correctly [21]. The underlying causes of death were recorded in the public-use linked mortality file using the following ICD-10 codes: cardiovascular diseases including heart diseases (I00–I09, I11, I13, I20–I51) and cerebrovascular diseases (I60–I69) and malignant neoplasms (C00-C97). From the 16,040 participants with baseline data, 1729 had missing data on mortality and other covariates. The final study sample included 14,311 participants.

### 2.3. Measurements of Blood Lead and Urinary Cadmium

Blood and urine samples were collected by the CDC during the medical examination. The laboratory methods for processing these samples are described in detail elsewhere [22]. Briefly, the blood and urine specimens were frozen (−30 and −20 °C, respectively), stored, and shipped for analysis to the Division of Laboratory Sciences, National Center for Environmental Health (Centers for Disease Control and Prevention in Atlanta, GA, USA). Lead (µg/dL) concentration was measured in whole blood using inductively coupled plasma mass spectrometry. Urinary cadmium was measured by graphite furnace atomic absorption with Zeeman background correction using the CDC method [22] with the modification proposed by Pruszkowska et al. [23]. Specimens were analyzed in duplicate and the average of the two measurements was reported. The detection limits were 1.0 μg/dL (0.048 μmol/L) and 0.03 μg/L for blood lead and urinary cadmium, respectively. For study participants who had concentrations of blood lead below the level of detection (*n* = 1217; 8.5%), values were imputed using LOD/√2 [0.7 μg/dL]. Urinary creatinine measured using the Jaffe reaction with a Beckman Synchron AS/ASTRA Clinical Analyzer (Beckman Instruments, Inc., Brea, CA, USA) was used to account for urine dilution.

### 2.4. Covariates

Baseline covariates were collected when individuals participated in a household interview and demographic information—including sex (male/female), age (continuous; years), race-ethnicity (non-Hispanic whites, non-Hispanic blacks, Mexican Americans, and others), poverty-to-income ratio (categorized in tertiles), the number of years of education attended and completed (continuous; years), area of residence (metro and non-metro counties), and smoking status (current, former, and never). Information on body-mass index ([BMI] continuous; kg/m [2]), physical activity (none, 1 to 14 times, 15 or more times; per month), and overall dietary quality indexes (continuous) was obtained during the medical examination. Dietary intake was collected using a 24-h dietary recall. We derived the diet quality indexes, as measured by the Healthy Eating Index 2015 (HEI-2015) [24] and the adapted dietary inflammatory index [25], from the daily intakes of foods/beverages, energy, and nutrients based on the 24-h dietary recall. A complete description of the development of these scores is described in Appendix A.

### 2.5. Statistical Analysis

Complete data on exposures, covariates, and mortality were available for 14,311 participants. We log-transformed (base 2) blood lead and urinary cadmium concentrations to reduce the influence of outliers, and descriptive and bivariate analyses are reported as geometric means and geometric standard errors (SE) by population characteristics. We used the parametric g-computation to estimate the risk ratio (RR) and risk differences (RD) of all-cause and specific causes of mortality under hypothetical interventions. In 1986, Robins introduced g-methods, a class of causal inference techniques that allows building an outcome prediction model based on observed quantities, and then predicts potential outcomes under potential hypothetical intervention [26,27]. In recent years, there have been substantial advances in the application of this method, which has been used, for instance, to evaluate hypothetical interventions for sources of lead exposure on BLLs [28]. The parametric g-formula is a generalization of the standardization method that enables flexible simulation and estimation of survival curves to visualize the time-specific effect estimates of any form of hypothetical intervention. A more detailed discussion of this method is presented elsewhere [29,30,31]. Briefly, we first estimated the expected mortality for all observations using confounders and metal exposure levels as predictors in a pooled logistic regression. The discrete-time hazards of all-cause and specific causes of mortality for each 2-fold increase in the baseline metals concentrations (log_2_-transformed to reduce skewness) were then estimated. Then, we used the model fit to predict every participant’s mortality while manipulating the exposure by setting it to (1) “high level” metal concentrations for every participant and (2) “low level” metal concentrations for every participant. Finally, we estimated the average difference between the expected mortality. We used the non-parametric bootstrap method (M = 200) to calculate the confidence intervals around the estimate (the 2.5th and 97.5th percentiles as the lower and upper confidence interval limits, respectively).

There are no known safety levels for the blood and urinary concentrations of these metals in adults. Therefore, we chose the interventions listed below based on previous epidemiologic analyses, as discussed elsewhere [14]. We compared the estimated risk of mortality under the following interventions: (1) all participants were assigned a high concentration (e.g., 95th percentile values of the metal distributions) with the estimated risk of mortality under the intervention. (2) All participants were assigned a low concentration (e.g., 5th percentile values of the metal distributions). This approach assumed a linear association between metal concentration and death. To check this assumption, we used multivariable restricted cubic splines with three knots placed at the 5th, 50th, and 95th percentiles of each metal concentration distribution to provide a graphical presentation [32]. Splines allowed us to test whether there was any departure from linearity.

Finally, we also considered interventions comparing quartile groups of lead and cadmium concentrations. We categorized metal concentrations into quartiles and estimated the discrete-time hazards of all-cause and specific causes of mortality for each quartile with the first quartile group, the lowest metal concentration, as the reference category. We then compared the estimated risk of mortality under the following interventions: (1) all participants belonged to the bottom quartile group with the estimated risk of mortality under the intervention. (2) All participants belonged to the low quartile group.

Models were adjusted for age, sex, ethnicity, poverty index, education level, area of residence, smoking status, BMI, physical activity, diet quality evaluated by the healthy eating index, and metal concentrations (mutual adjustment). The selection of potential confounders was done a priori. We also included product terms between the metal concentrations and time in all models to account for the time-varying risk. All analyses were weighted by the provided sample weights to account for the unequal probabilities of inclusion and response rates. 

Additionally, we investigated age (<50 years and ≥50 years) and sex-specific estimates in stratified analyses since previous studies reported potential effect modifications of the associations between metal concentrations and mortality by sex and age [12,14]. We also applied the Wald test [33] to assess the difference in associations of metal exposures with risk of mortality between subgroups of sex and age. A *p*-value < 0.1 was considered to be statistically significant.

### 2.6. Sensitivity Analyses

We ran sensitivity analyses using unweighted models that did not account for unequal probabilities of inclusion and response rates in the NHANES survey because the weighted method was inefficient due to the large variability in assigned weights [34]. Unweighted analysis yields correct estimates when models are adjusted for the auxiliary variables used to define the weights (i.e., age, sex, and ethnicity) [34]. Finally, as smoking is a source of lead and cadmium, the main analysis was adjusted for exposure to tobacco smoke measured by cotinine level and pack-years of cumulative active smoking to further account for residual confounding by smoking.

Statistical analyses were performed using R version 4.0.4 and Statistical Analysis System software version 9.4 (SAS Institute, Cary, NC, USA).

## 3. Results

A total of 14,311 participants were included (mean age 48.0 ± 18.1 years) for this analysis. The blood lead concentrations ranged from 0.70 to 56.0 μg/dL (0.034 to 2.70 μmol/L), with a geometric mean (GM) of 2.97 μg/dL (geometric standard error [GSE]= 1.01). BLLs were higher in older participants, males, current and former smokers, those who reported drinking alcohol more than four time per month, and those who were in the categories of lower healthy eating index and low poverty-to-income ratio (Table 1). The urinary creatinine-standardized cadmium concentrations ranged from 0.002 to 23.35 μg/g, with a GM of 0.36 μg/g (GSE = 1.01). Participants who had the highest concentrations of urinary cadmium were older and more likely to be female, current and former smokers, and to not practice physical activity. There were no major differences among other participant characteristics (Table 1).

### 3.1. Blood Lead and Urinary Cadmium Concentrations and All-Cause and Cause-Specific Mortality

During a median follow-up of 22.5 years (IQR 16.3–24.7), 5167 (36%) participants died with 1550 (30%) and 1135 (22%) deaths attributable to cardiovascular disease and cancer, respectively. 

When comparing a potential intervention assigning all participants to 5th percentile values (0.70 μg/dL and 0.04 μg/g for blood lead and urinary cadmium levels; respectively) to an intervention assigning all participants to 95th percentile values (9.70 μg/dL and 1.63 μg/g for blood lead and urinary cadmium levels; respectively), we observed 138% (95% CI, 14 to 196) and 126% (95% CI, 77 to 383) increases in the risk of all-cause mortality at 27 years of follow-up for blood lead and urinary cadmium, respectively. On the absolute scale, we observed 4.17% (95% CI, 1.54 to 8.77) and 6.22% (95% CI, 4.51 to 12.00) increases in all-cause mortality associated with blood lead and urinary cadmium, respectively (Figure 1, Table 2). 

For cause-specific mortality, comparing the 5th to the 95th percentile assignments, blood lead and urinary cadmium showed a 109% (95% CI, 4 to 268) and 37% (95% CI, −0.20 to 239) increased risk of cardiovascular disease mortality at 27 years of follow-up, respectively (Figure 1, Table 2). The estimated 27-year cancer mortality risk after intervention to set all participants to the 95th percentile compared to the 5th percentile was increased by 287% (95% CI, 12 to 691) and 144% (95% CI, −0.34 to 281) for blood lead and urinary cadmium, respectively (Figure 1, Table 2). We did not observe any departure from linearity in the associations between metal concentrations and mortality when using smoothing splines (Appendix A).

When we considered interventions comparing quartile groups of blood lead and urinary cadmium levels, the estimated RD of 27-year all-cause mortality after intervening to set all participants to the fourth quartile of metal concentrations compared to setting all participants to the first quartile of metal concentrations showed a percentage point increase of 5.52% (95% CI, 1.25 to 18.25) and 10.48% (95% CI, 5.15 to 27.99) for blood lead and urinary cadmium, respectively (Figure 2, Table 3). The corresponding increases in the 27-year RD for cardiovascular disease mortality were 5.18% (95% CI, 0.47 to 48.61) and 6.31% (95% CI, 3.76 to 50.38) for blood lead and urinary cadmium, respectively. For cancer mortality, the 27-year RD increased 4.17% (95% CI, −0.28 to 26.00) and 0.13% (95% CI, −1.57 to 48.35) for blood lead and urinary cadmium, respectively (Figure 2, Table 3).

### 3.2. Analyses Stratified by Sex and Age

We found that the associations between the blood lead concentration and all-cause (p-heterogeneity = 0.021) and cardiovascular disease (p-heterogeneity <0.001) mortality were more pronounced for older (≥50 years) compared to younger (<50 years) participants, while for cancer mortality, the association was similar for younger and older participants (p-heterogeneity = 0.429) (Appendix A). The associations between the urinary cadmium concentration and all-cause (p-heterogeneity = 0.020) and cardiovascular disease (p-heterogeneity = 0.093) mortality were more pronounced for older (≥50 years) compared to younger (<50 years) participants, while for cancer mortality, the association was similar for younger and older participants (p-heterogeneity = 0.525) (Appendix A). Analyses stratified by sex showed that the associations between the urinary cadmium concentration and all-cause (p-heterogeneity = 0.098), cardiovascular disease (p-heterogeneity = 0.032), and cancer (p-heterogeneity = 0.102) mortality were more pronounced in men. There was no difference in the mortality risks associated with blood lead concentration and all-cause (p-heterogeneity = 0.622), cardiovascular disease (p-heterogeneity = 0.202), and cancer (p-heterogeneity = 0.206) mortality (Appendix A).

### 3.3. Sensitivity Analyses

When the models were unweighted, we observed the same pattern of associations for all-cause and cancer mortality both for lead and cadmium exposures (Appendix A). However, the association between blood lead concentration and cardiovascular disease mortality was more pronounced with an RR of 2.09 (95% CI, 1.04 to 3.68) at 27 years of follow-up compared to the model without the survey weights (RR = 1.02 (0.52 to 2.80), which corresponded to a percentage point increase of 0.69% (95% CI, −2.72 to 3.85) (Appendix A). The model with the adjustment for more precise tobacco smoking measurement showed the same pattern of associations (Appendix A).

## 4. Discussion

Using a large, nationally representative sample of US adults, we found that in those with high concentrations of blood lead and urinary cadmium, there were excess deaths from all causes (corresponding to 6,460,092 and 9,635,916 deaths, respectively, when considering NHANES weights), cardiovascular disease (corresponding to 2,354,758 and 1,642,134 deaths, respectively), and cancer (corresponding to 2,044,921 and 991,477 deaths, respectively) after 27 years. The associations were more pronounced for older participants, except for cancer mortality associated with blood lead levels. In addition, all of these associations were more pronounced in men than in women for urinary cadmium concentrations.

Our findings are in line with previous population-based studies showing that exposures to lead and cadmium were associated with increased risks of cardiovascular, cancer, and all-cause mortality [12,13,14,15,16]. A recent study showed that environmental declines in lead and cadmium exposures were associated with a 32% reduction in cardiovascular disease mortality [35]. Two studies reported that exposure to chronic, low concentrations of lead was associated with premature death, especially from cardiovascular disease [14,15]. In contrast, there was no association with cancer mortality risk [15]. Several epidemiological studies have prospectively examined the associations of cadmium exposure in relation to the risk of all-cause [12,36,37,38,39,40,41], cancer [37,42,43,44], and cardiovascular disease mortality [12,37,38,39,42]. Most studies have reported a positive association between urinary cadmium concentrations and mortality, except one study for cardiovascular disease mortality [12], two studies for cancer mortality [37,42] and two studies for all-cause mortality [12,36]. In comparison to other studies, our study explored the potential effect on mortality for different theoretical interventions of exposure levels of these metals.

Environmental exposure to lead occurs continuously over a lifetime and lead is retained in the body for decades. Blood lead is an established biomarker of recent exposure, although it also shows a small component with a half-life of 5–20 years that reflects endogenous exposure from bone lead redistribution [45]. Several mechanisms have been proposed for the role of lead in cardiovascular events. Lead exposure can result in oxidative stress, inflammation, and diminished endothelium relaxation, and it promotes the development of atherosclerosis and thrombosis [46]. In addition, lead is a well-known risk factor for hypertension and has been associated with peripheral arterial disease, electrocardiographic abnormalities, left-ventricular hypertrophy, alteration of cardiac conduction, cardiac disease, and increased mortality due to cardiovascular disease [45,47]. Regarding cancer toxicity, the mechanisms by which lead may lead to tumor development is unclear. However, lead has been defined as a “probable human carcinogen” by the International Agency for Research on Cancer [48]. It has been proposed that lead can facilitate the process of carcinogenesis by inhibiting DNA synthesis and repair and by interacting with binding proteins, thus hindering tumor suppressor proteins [49]. Lead may also affect carcinogenesis through mechanisms involving oxidative damage, induction of apoptosis, and altered signaling pathways [50].

While blood cadmium tends to reflect recent exposures, urinary cadmium reflects kidney cadmium contents and, with a half-life of 15–30 years, is an established biomarker of cumulative body burden. The cardiovascular toxic effects of cadmium exposure have been well described. Experimental evidence supports a role for cadmium in atherosclerosis, including increased inflammation [51] and endothelial oxidative stress [52,53]. Results from epidemiological studies show that high cadmium exposure is associated with hypertension [54], growth of atherosclerotic plaques [55,56] and cardiovascular disease [57,58,59]. As mentioned recently by Lamas et al. [60], strong evidence supports that it is time to recognize metal contaminants in the evaluation, treatment, and prevention of cardiovascular diseases. Evidence for cancer toxicity stems from various mechanisms. Cadmium may promote carcinogenesis through induction of oxidative stress [61,62], suppression of DNA, and changes in DNA methylation and apoptosis repair [62,63,64]. Another alternative hypothesis is through estrogenic activities [65].

This study used the NHANES III dataset, a large, national survey of which the findings are generalizable to the adult, non-institutionalized, U.S. population. There are many strengths to this study, including its large sample-size and random sampling, the mutual adjustment between lead and cadmium metals, the adjustment for dietary intake (one of the main sources of exposure) and, most importantly, we used a method that was not based on proportional hazard assumptions and the graphic representation of the risk at each follow-up time allowed us to show that the risk varied over the duration of the follow-up. For example, the risk differences for cardiovascular disease mortality associated with blood lead and urinary cadmium concentrations were +1.52% and +1.06% at the end of the follow-up and +0.47% and +0.27% at the mid-follow-up (13 year), respectively (Appendix A). Nevertheless, there are important limitations to note. The key limitation is that we relied on blood concentrations for lead; therefore, we did not account for cumulative chronic or long-term lead exposure. Urine measurements for lead are better indicators of cumulative exposure and would have strengthened the results. Furthermore, covariable data were only available at baseline. Thus, exposure-confounder relationships were not well defined temporally. Another limitation is that we relied on death certificates for the underlying cause of death, but they are imperfect [66]. Most importantly, although the main potential confounding factors were accounted for, there could be other cardiovascular disease and cancer risk factors and unmeasured confounding from other toxicants, especially in occupational settings, which may have influenced our findings. Finally, because internal dose metrics cannot correctly define the complete history of exposure and duration, the timing that correlates most strongly with the observed health effects are typically unknown or highly uncertain.

This study focused only on hypothetical interventions related to lead and cadmium exposures to simulate what would have been the benefits of historical interventions. We recommend that future directions in environmental health research explore other interventions based on, for example, dietary and lifestyle factors, which may be complementary to pollutant-based interventions.

In conclusion, our findings suggest that blood lead and urinary cadmium are associated with all-cause, cardiovascular disease, and cancer mortality. Despite the continuous decrease in lead and cadmium exposures in the U.S. population, we confirmed the previously reported associations and showed that several deaths could have been prevented under interventions to reduce the blood lead concentration from 9.70 to 0.70 μg/dL and the creatinine-corrected urinary cadmium concentration from 1.63 to 0.04 μg/g. Interventions to reduce historical exposures to these metals would be more effective (on the absolute scale) among individuals ≥50 years of age as well as in men.

## Figures and Tables

**Figure 1 toxics-11-00133-f001:**
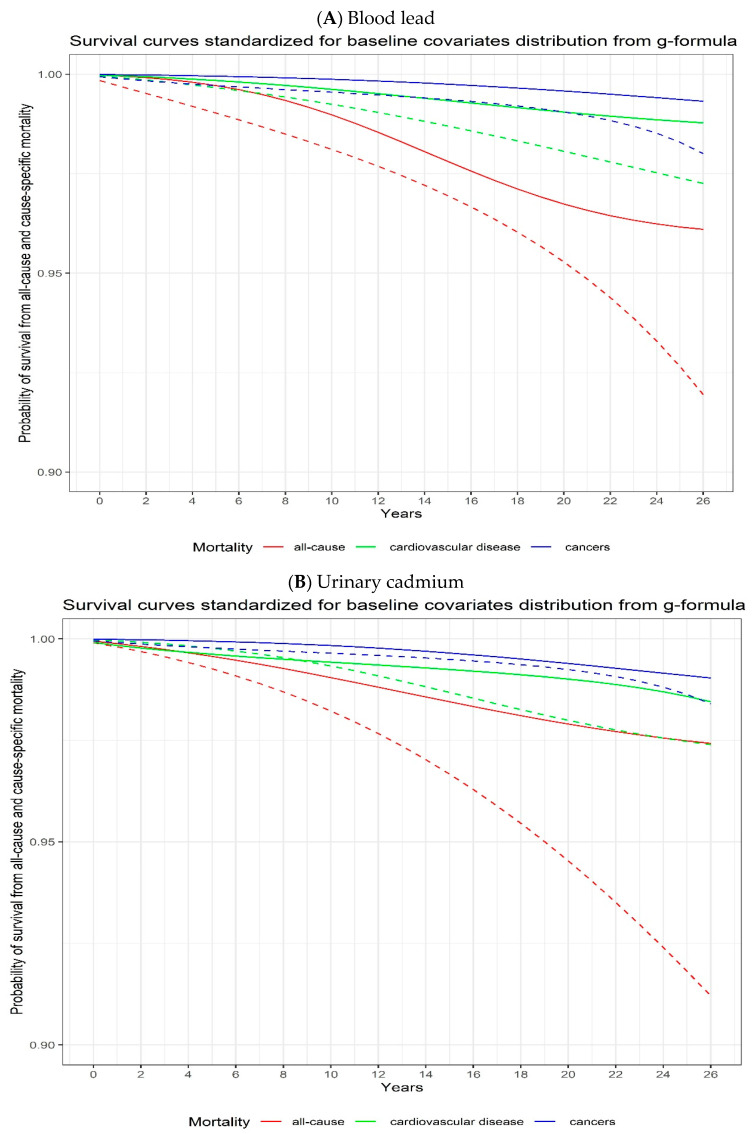
Adjusted all-cause, cardiovascular disease, and cancer mortality risks according to blood lead (**A**) and urinary cadmium (**B**) levels using parametric g-formula with pooled logistic regression models. The solid lines represent the 5th percentile (0.70 μg/dL for lead and 0.04 μg/L for cadmium) and the dashed lines represent the 95th percentile (9.70 μg/dL for lead and 1.63 μg/L for cadmium). Robust 95% confidence intervals (CIs) for each exposure category estimated by bootstrapping (in the pooled logistic regression model) are presented in Table 2.

**Figure 2 toxics-11-00133-f002:**
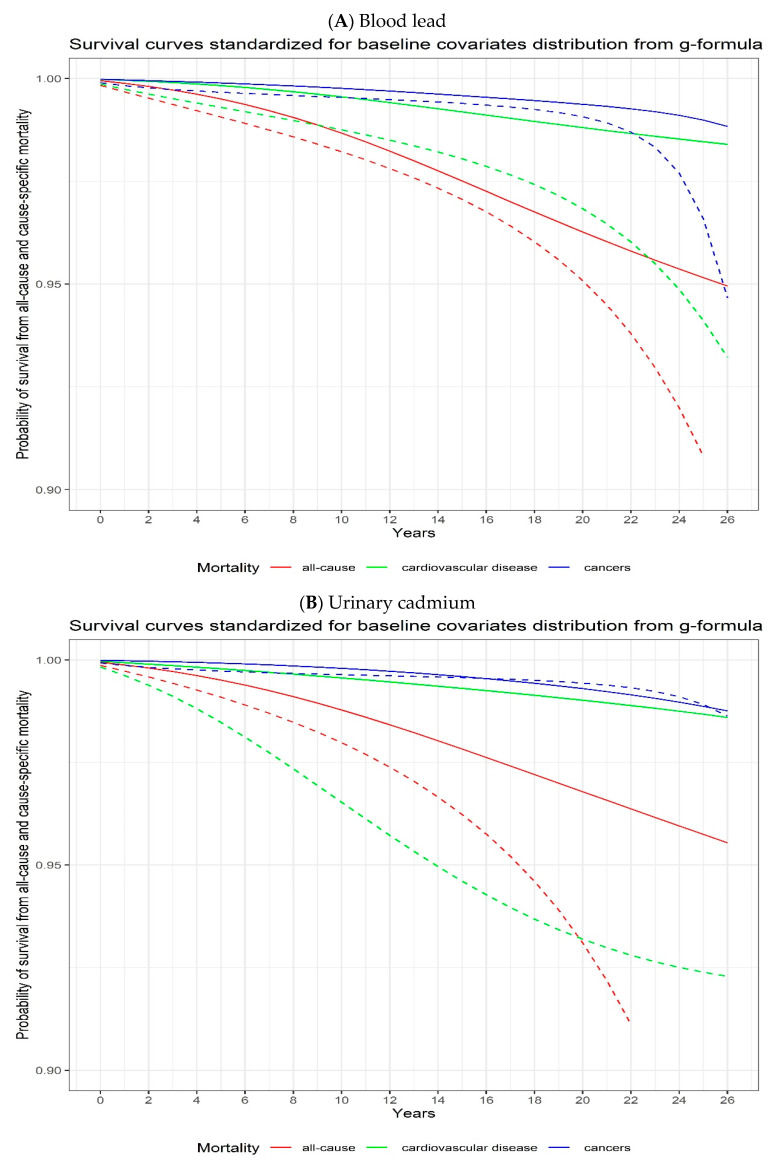
Adjusted all-cause, cardiovascular disease, and cancer mortality risks according to blood lead (**A**) and urinary cadmium (**B**) levels using parametric g-formula with pooled logistic regression models. The solid lines represent the first quartile and the dashed lines represent the fourth quartile. Robust 95% confidence intervals (CIs) for each exposure category estimated by bootstrapping (in the pooled logistic regression model) are presented in Table 3.

**Table 1 toxics-11-00133-t001:** Geometric means and standard errors of blood lead and urinary cadmium levels by study population baseline characteristics.

		Blood Lead (μg/dL)	Urinary Cadmium (μg/g)
	N (%)	GM (GSE)	GM (GSE)
Age classes (years)			
<50	8170 (57.09)	2.59 (1.01)	0.26 (1.01)
≥50	6141 (42.91)	3.56 (1.01)	0.56 (1.01)
Sex			
Women	7570 (52.90)	2.26 (1.01)	0.42 (1.01)
Men	6741 (47.10)	4.03 (1.01)	0.31 (1.01)
Ethnicity			
Mexican-American	3891 (27.19)	2.99 (1.01)	0.31 (1.02)
Other Hispanic	369 (2.58)	2.88 (1.04)	0.32 (1.07)
Non-Hispanic	10,051 (70.23)	2.96 (1.01)	0.39 (1.01)
Smoking status			
Never smoker	6807 (47.7)	2.37 (1.01)	0.27 (1.01)
Current smoker	4477 (31.28)	3.86 (1.01)	0.48 (1.02)
Former smoker	3027 (21.15)	3.32 (1.01)	0.46 (1.02)
Body-mass index			
Normal weight (<25·0 kg/m^2^)	5707 (39.88)	2.91 (1.01)	0.34 (1.02)
Overweight (25·0–29·9 kg/m^2^)	4855 (33.92)	3.18 (1.01)	0.38 (1.02)
Obese (≥30·0 kg/m^2^)	3749 (26.20)	2.80 (1.01)	0.38 (1.02)
Physical activity (per month)			
None	2939 (20.54)	3.14 (1.01)	0.44 (1.02)
One to 14 times	5099 (35.63)	2.89 (1.01)	0.37 (1.02)
15 or more times	6273 (43.83)	2.95 (1.01)	0.32 (1.02)
Poverty to income ratio tertile			
First	4776 (33.37)	3.21 (1.01)	0.38 (1.02)
Second	4765 (33.30)	2.85 (1.01)	0.36 (1.02)
Third	4770 (33.33)	2.85 (1.01)	0.35 (1.02)
Alcohol intake (drinks per month)			
Four or fewer	9940 (69.46)	2.72 (1.01)	0.38 (1.01)
More than four	4371 (30.54)	3.63 (1.01)	0.33 (1.02)
Healthy eating index tertile			
First	4765 (33.30)	3.23 (1.01)	0.36 (1.02)
Second	4774 (33.36)	2.91 (1.01)	0.35 (1.02)
Third	4772 (33.34)	2.78 (1.01)	0.38 (1.02)
Adapted dietary inflammatory index tertile			
First	4770 (0.33)	2.79 (1.01)	0.34 (1.02)
Second	4771 (0.33)	2.95 (1.01)	0.35 (1.02)
Third	4770 (0.33)	3.17 (1.01)	0.40 (1.02)

GM: geometric mean, GSE: geometric standard errors.

**Table 2 toxics-11-00133-t002:** Adjusted risk ratio and risk difference of all-cause and specific-cause of mortality for interventions assigning 95th percentile values compared to 5th percentile values of metal concentrations using parametric g-formula with pooled logistic regression models ^a^.

Outcomes	Blood Lead	Urinary Cadmium
All-cause Mortality		
Number of events (%) among low level before the intervention	177/1217 (14.5%)	122/682 (17.9%)
Number of events (%) among high level before the intervention	4767/13,603 (35.0%)	4645/13,598 (34.2%)
Adjusted risk ratio (95% CI)	2.38 (1.14 to 2.96)	2.26 (1.77 to 4.83)
Adjusted risk difference (95% CI)	+4.17% (1.54 to 8.77)	+6.22% (4.51 to 12.00)
Cardiovascular disease mortality		
Number of events (%) among low level before the intervention	35/1217 (2.9%)	30/682 (4.4%)
Number of events (%) among high level before the intervention	1423/13,603 (10.5%)	1405/13,598 (10.3%)
Adjusted risk ratio (95% CI)	2.09 (1.04 to 3.68)	1.37 (0.80 to 3.39)
Adjusted risk difference (95% CI)	+1.52% (0.09 to 3.74)	+1.06% (−0.57 to 3.50)
Cancer mortality		
Number of events (%) among low level before the intervention	49/1217 (4.0%)	22/682 (3.2%)
Number of events (%) among high level before the intervention	1027/13,603 (7.6%)	1007/13,598 (7.4%)
Adjusted risk ratio (95% CI)	3.87 (1.12 to 7.91)	2.44 (0.66 to 3.81)
Adjusted risk difference (95% CI)	+1.32% (−0.09 to 3.67)	+0.64% (−0.98 to 2.80)

^a^ 200 iterations were performed for bootstrapping the estimates at 95% confidence interval; low level: 5th percentiles of blood lead (0.70 μg/dL or 0.03 umol/L) and urinary cadmium (0.04 μg/g) distributions; high level: 95th percentiles of blood lead (9.70 μg/dL or 0.47 umol/L) and urinary cadmium (1.63 μg/g) distributions.

**Table 3 toxics-11-00133-t003:** Adjusted risk ratio and risk difference of all-cause and specific-cause of mortality for interventions setting participants in the fourth quartile compared to the first quartile for metal concentrations using parametric g-formula with pooled logistic regression models ^a^.

Outcomes	Blood Lead	Urinary Cadmium
All-cause mortality		
Number of events (%) among quartile 1 before the intervention	695/3640 (19.1%)	504/3578 (14.1%)
Number of events (%) among quartile 4 before the intervention	1877/3564 (52.7%)	2157/3578 (60.3%)
Adjusted risk ratio (95% CI)	1.61 (1.12 to 3.34)	2.09 (1.75 to 5.42)
Adjusted risk difference (95% CI)	+5.52% (1.25 to 18.25)	+10.48% (5.15 to 27.99)
Cardiovascular disease mortality		
Number of events (%) among quartile 1 before the intervention	168/3640 (4.6%)	123/3578 (3.4%)
Number of events (%) among quartile 4 before the intervention	589/3564 (16.5%)	631/3578 (17.6%)
Adjusted risk ratio (95% CI)	3.38 (1.29 to 25.66)	7.18 (3.12 to 78.03)
Adjusted risk difference (95% CI)	+5.18% (0.47 to 48.61)	+6.31% (3.76 to 50.38)
Cancer mortality		
Number of events (%) among quartile 1 before the intervention	172/3640 (4.7%)	98/3578 (2.7%)
Number of events (%) among quartile 4 before the intervention	431/3564 (12.1%)	544/3578 (15.2%)
Adjusted risk ratio (95% CI)	2.44 (1.01 to 13.48)	2.08 (0.42 to 40.73)
Adjusted risk difference (95% CI)	+4.17% (−0.28 to 26.00)	+0.13% (−1.57 to 48.35)

^a^ 200 iterations were performed for bootstrapping the estimates at 95% confidence interval.

## Data Availability

All the data are available on the Centers for Disease Control and Prevention website https://www.cdc.gov/nchs/nhanes/index.htm, accessed on 12 March 2021.

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
