# Peer review of "Associations with Blood Lead and Urinary Cadmium Concentrations in Relation to Mortality in the US Population: A Causal Survival Analysis with G-Computation"

_toxics, 2023, doi:10.3390/toxics11020133_

Round 1
Reviewer 1 Report
The manuscript evaluated the associations of blood lead and urinary cadmium concentrations in relation to mortality in the US population, based on data from the NHANES-III, using g-parametric formula.
The manuscript is interesting, taking into consideration that cancer and cardiovascular disease (CVD) are the two most common causes of mortality and morbidity worldwide. Although commonly thought of as 2 separate disease entities, CVD and cancer possess various similarities and possible interactions, including a number of similar risk factors, suggesting a shared biology for which there is emerging evidence. Although chronic inflammation is an indispensable feature of the pathogenesis and progression of both CVD and cancer, additional mechanisms can be found at their intersection.
Among the risk factors, the authors discussed the Cd and Pb levels in relation with both diseases, considering that both metals are chemicals listed by the World Health Organization as environmental pollutants of major public health concern.
The manuscript is robust, well organized and used a large battery of data. The experiments were properly performed and the results correctly interpreted and discussed.
Author Response
Referee 1
The manuscript evaluated the associations of blood lead and urinary cadmium concentrations in relation to mortality in the US population, based on data from the NHANES-III, using g-parametric formula.
The manuscript is interesting, taking into consideration that cancer and cardiovascular disease (CVD) are the two most common causes of mortality and morbidity worldwide. Although commonly thought of as 2 separate disease entities, CVD and cancer possess various similarities and possible interactions, including a number of similar risk factors, suggesting a shared biology for which there is emerging evidence. Although chronic inflammation is an indispensable feature of the pathogenesis and progression of both CVD and cancer, additional mechanisms can be found at their intersection.
Among the risk factors, the authors discussed the Cd and Pb levels in relation with both diseases, considering that both metals are chemicals listed by the World Health Organization as environmental pollutants of major public health concern.
The manuscript is robust, well organized and used a large battery of data. The experiments were properly performed and the results correctly interpreted and discussed.
Response: We would like to thank the reviewer for their time and comments.
Reviewer 2 Report
The topic of the mnauscript fits into de scope of the journal. Lead and cadmium has historically associated with increased risks of cardiovascular, cancer, and all-cause mortality. Therefore, any additional and novel approach and formula to fill the still existing knowledge gaps is of great interest.
The paper has been well designed and written.
The objectives are clear and the statistical method has been well described.
The authors have not generated new data but use the data from the US National Health and Nutrition Examination Survey (NHANES)
The conclusions suggest future interventions although nothing new not previously known or discussed by other authors.
The description of the method may generate some misundertanding related to the execution of the analysis and data collection.
It is not clear if the section "Measurements of blood lead and urinary cadmium" was executed by the authors or by the NHANES III survey in 2015?
The same with the section "The interview includes background information such as socio-demographic, dietary and health-related questions. The examination component consists of medical and physiological measurements, as well as laboratory tests". It is not clear if the authors executed these interviews or just reviewed the previously interviews executed by NHANES.
What were the authors tasks in the execution of the method?.
Author Response
Referee 2
The topic of the mnauscript fits into de scope of the journal. Lead and cadmium has historically associated with increased risks of cardiovascular, cancer, and all-cause mortality. Therefore, any additional and novel approach and formula to fill the still existing knowledge gaps is of great interest.
The paper has been well designed and written.
The objectives are clear and the statistical method has been well described.
The authors have not generated new data but use the data from the US National Health and Nutrition Examination Survey (NHANES)
Response: We would like to thank the reviewer for their time and comments. We think the comments have helped us improve our manuscript. Please find below our detailed response to the suggested comments.
The conclusions suggest future interventions although nothing new not previously known or discussed by other authors.
Response: Indeed, as perspectives, we suggested that future directions in environmental health research explore other interventions based on, for example, dietary and lifestyle factors, which may be complementary to pollutant-based interventions.
The description of the method may generate some misundertanding related to the execution of the analysis and data collection.
It is not clear if the section "Measurements of blood lead and urinary cadmium" was executed by the authors or by the NHANES III survey in 2015?
The same with the section "The interview includes background information such as socio-demographic, dietary and health-related questions. The examination component consists of medical and physiological measurements, as well as laboratory tests". It is not clear if the authors executed these interviews or just reviewed the previously interviews executed by NHANES.
Response: Thank you for this comment. We have clarified this point in the method section.
“The US National Health and Nutrition Examination Survey (NHANES) data, collected, stored, and analyzed by the Centers for Disease Control and Prevention (CDC) were used for this study”
What were the authors tasks in the execution of the method?.
Response: We provided the link to access the NHANES study protocol in the methods section. In addition to the protocol, there is the presentation of the data shared by the CDC. After downloading the data needed for our study, we performed all the statistical analyses, from data cleaning to the implementation of the statistical models.
Reviewer 3 Report
This paper is well developed with high quality analysis. A couple of minor revisions:
1. Authors should explain why ethnicity is classified as "Mexican-American, other Hispanic, not Hispanic", not "African American, the White and others"?
2. Some baseline covariates are self-reported, such as the smoking status. It might include some bias. Thus, it should be mentioned in the limitations.
Author Response
Referee 3
This paper is well developed with high quality analysis. A couple of minor revisions:
- Authors should explain why ethnicity is classified as "Mexican-American, other Hispanic, not Hispanic", not "African American, the White and others"?
Response: We would like to thank the reviewer for their time and comments. In NHANES, four race-ethnicity categories were used (also recommended for data analysis): non-Hispanic whites, non-Hispanic blacks, Mexican Americans, and others. We have now corrected this information.
“Baseline covariates were collected when individuals participated in a household interview and demographic information—including sex (male/female), age (continuous; years), race-ethnicity (non-Hispanic whites, non-Hispanic blacks, Mexican Americans, and others),..”
- Some baseline covariates are self-reported, such as the smoking status. It might include some bias. Thus, it should be mentioned in the limitations.
Response: We have reported this limitation in the discussion section.
“Furthermore, covariables data were only available at baseline. Thus, the exposure-confounders relationships were not well defined temporally.”
Reviewer 4 Report
This is an interesting study and worthy of eventual publication in Toxics. I have some minor issues.
I believe the GSE should be reported to two decimal places in table 1, as the GM are. This would probably eliminate so many zero entries, which are basically meaningless. I similar significant figure error appears in the text above table 1 (0.36 microg/g (GSE=0.003)).
The manuscript suffers from numerous grammatical and related errors. For example, the word "for" or something similar appears to be missing from the start of fourth sentence of the introduction. The sentence in lines 56 and 57 is actually a sentence fragment. In line 167, "safety" should be "safe".
The manuscript would read better if first person voice, i.e., "we", was limited to the introduction and conclusion.
Author Response
Referee 4
This is an interesting study and worthy of eventual publication in Toxics. I have some minor issues.
Response: We would like to thank the reviewer for their time and comments. We think the comments have helped us improve our manuscript. Please find below our detailed response to the suggested comments.
I believe the GSE should be reported to two decimal places in table 1, as the GM are. This would probably eliminate so many zero entries, which are basically meaningless. I similar significant figure error appears in the text above table 1 (0.36 microg/g (GSE=0.003)).
Response: Thank you for the suggestion, we now reported GSE with two decimals.
The manuscript suffers from numerous grammatical and related errors. For example, the word "for" or something similar appears to be missing from the start of fourth sentence of the introduction. The sentence in lines 56 and 57 is actually a sentence fragment. In line 167, "safety" should be "safe".
Response: Thank you for this comment. We have corrected the corresponding sentences and proofread the paper.
The manuscript would read better if first person voice, i.e., "we", was limited to the introduction and conclusion.
Response: Thank you for the suggestion. We modified the method section accordingly.
Round 2
Reviewer 2 Report
thanks for addressing all comments